# Mortality among People with Epilepsy: A Retrospective Nationwide Analysis from 2016 to 2019

**DOI:** 10.3390/ijerph181910512

**Published:** 2021-10-07

**Authors:** Kristijonas Puteikis, Rūta Mameniškienė

**Affiliations:** 1Faculty of Medicine, Vilnius University, 03101 Vilnius, Lithuania; kristijonas.puteikis@mf.stud.vu.lt; 2Center for Neurology, Vilnius University, 08661 Vilnius, Lithuania

**Keywords:** premature death, seizures, sudden unexpected death in epilepsy, suicide

## Abstract

We estimated age-adjusted mortality and investigated the dominant causes of death as well as comorbidities among people with epilepsy (PWE) in Lithuania, a country with frequent deaths from external causes. From 2016 to 2019, the age-adjusted rate of death among PWE in Lithuania was compared with mortality data in the general population. Each year of analysis, individuals who were diagnosed with epilepsy comprised a retrospective cohort. The standardized mortality ratio (SMR) of PWE varied from 2.93 (95% CI 2.78 to 3.07) to 3.18 (95% CI 3.02 to 3.34). PWE died at least one decade earlier than expected in the general population. The dominant causes of death were cardiovascular diseases (their proportion ranged from 44.8% to 49.3%), cancer (16.7% to 21.3%) and external causes of death (8.5% to 10.9%). The proportion of the latter decreased over time (r = −0.99, *p* = 0.01), whereas the SMR for external causes of death remained relatively constant. Epilepsy was the underlying cause of death in 163 cases (2.6%), and noted as a condition contributing to death in 1010 cases (15.9%). Cerebrovascular and cardiological conditions and dementia were the most frequent comorbidities among PWE before their death. Epilepsy-unrelated causes of death are relevant contributors to mortality among PWE. There is a need for PWE-oriented societal interventions to reduce the frequency of external deaths beyond the trend in the general population.

## 1. Introduction

Epilepsy is a burdensome neurological disease known to increase mortality by up to 10 times among afflicted persons [1,2]. As secondary causes of epilepsy and co-existent neurological diseases become more prevalent, ageing populations are expected to face a rise in epilepsy-related mortality [3].

The risks that emerge after epilepsy is diagnosed are multifactorial—seizures may cause sudden unexpected death in epilepsy (SUDEP); lead to falls or drowning; death can occur because of underlying causes of the disorder (e.g., cerebrovascular diseases) or epilepsy-associated mental health issues that lead to suicide [4]. Such deaths may be termed as “epilepsy-related”. A large sphere of mortality research in epileptology has centered around a subgroup of epilepsy-related deaths—those that are directly caused by seizures. The possibilities of preventing dramatic occurrences of SUDEP have received attention from epilepsy specialists as well as patients, their caregivers and patient organizations [5]. However, incomplete knowledge of the phenomenon, difficulties of documenting cases of SUDEP in routine healthcare records, and the lack of SUDEP-specific prevention call for the additional research of deaths among people with epilepsy (PWE) [6,7]. Recent findings signal that preventing deaths due to external causes, such as suicide, unintentional injury or homicide, could be one of the priorities when seeking to improve survival in this patient group [8,9]. Pharmacological and surgical efforts dedicated to achieving seizure freedom may decrease the risk of SUDEP, but additional measures (e.g., psychological consultations) could supplement care for PWE that are not adherent to treatment or have treatment-resistant epilepsy [10]. Therefore, it is important to better understand the main preventable causes of death beyond SUDEP.

Lithuania is the southernmost of the three Baltic states, and currently a member of the European Union. The country saw some of the highest rates of death because of external causes throughout the last decades and has become a case-study of how complex societal and economical changes have led to substantial increases in suicides, predominantly in the male population [11,12]. The region, however, is under-represented in the context of international research of mortality in epilepsy and may provide novel information about deaths from external causes in epilepsy [13].

The aims of the current study were (1) to determine the age-standardized mortality ratio (SMR) among PWE in Lithuania each year from 2016 to 2019, and (2) to investigate dominant causes of death and cause-specific mortality during this period.

## 2. Materials and Methods

### 2.1. Study Design and Data Sources

We conducted a retrospective cohort study, in which the age-adjusted rates of death among PWE in 2016, 2017, 2018 and 2019 were compared with mortality in the general population of Lithuania during the respective years. Individuals with epilepsy were identified from records of an anonymized copy of the database of the National Compulsory Health Insurance Fund (NCHIF), which consists of medical records of all permanent residents in Lithuania (they are covered by the NCHIF through compulsory contributions). The inclusion of PWE each year relied on a documented diagnosis of epilepsy (epilepsy, G40 (both with or without status epilepticus, G41), according to the International Statistical Classification of Diseases and Related Health Problems, Tenth Revision, Australian Modification [ICD-10-AM]) either the year of analysis and/or anytime two years before (Figure 1). The age and sex of each person with epilepsy were extracted. The anonymized nature of the dataset did not allow us to perform validation studies. However, it has been shown that including individuals with codes G40 and G41 provides adequate sensitivity and specificity when investigating mortality among PWE [14,15]. The diagnosis of epilepsy in Lithuania is registered in electronic records every time a person with epilepsy has been consulted by a healthcare provider and/or prescribed antiseizure drugs (ASDs) for their disorder. Therefore, the use of ASDs, which is recommended to be included in studies prioritizing high positive predictive value, was indirectly reflected by a repeated documentation of the G40 diagnosis. The latter ICD-10-AM code is entered into the system of the NCHIF at each visit and ASDs cannot be prescribed for a period exceeding six months; therefore, the two-year window for G40 to be recorded before the year of analysis was judged to be unrestrictive.

Individual deaths among PWE each year were identified by joining the database of the NCHIF with records of The State Register of Death Cases and Their Causes, which relies on death certificates to collect death-related data. From 2016 to 2019, there were 8, 11, 10 and 15 deceased PWE included in the NCHIF database, respectively, who could not be found in the register, and were therefore omitted from further analyses. For all deceased PWE, age at death, sex, underlying causes of death and other significant conditions contributing to death were extracted (these data originated from death certificates). All co-existing comorbidities (in the form of ICD-10-AM codes) documented the year of death or two years before were extracted as well (they originated from the NCHIF database). Comorbidities that had been present among the deceased individuals were grouped according to components of the epilepsy-specific comorbidity index developed by St. Germaine-Smith et al. (supplemented by the group of cerebrovascular diseases) [16].

All data regarding PWE were provided by the Institute of Hygiene of Lithuania, an institution functioning under the Ministry of Health that manages public-health-related data. The age structure of the general population of Lithuania as well as the number of cause-specific deaths in selected age groups originated from open portals of the Lithuanian Department of Statistics and the National Institute of Hygiene of Lithuania [17,18].

Summarized characteristics of the dataset used in the study are presented in Table 1.

### 2.2. Data Analysis

Statistical analyses were performed in Microsoft Excel 2016 and IBM SPSS v26. The normality of the age distribution among selected subgroups was assessed by observing Q–Q plots and performing the Kolmogorov–Smirnov test. The non-parametric Kruskal–Wallis test was used to detect differences between the median age at death in different years. It was selected because of a non-normal age distribution and the availability of post hoc comparisons of age between different years of analysis. Pearson’s correlation coefficients were calculated to detect trends in the percentage of deaths from certain causes throughout the years.

The numbers of expected deaths and the standardized mortality ratios among people with epilepsy were calculated by means of indirect standardization [19]. The reference population consisted of all inhabitants of Lithuania at the start of the year of analysis. The age structure was calculated every five years for all-cause mortality and every ten years for cause-specific mortality.

Potential years of life lost (PYLL) were calculated with the reference age set at 75 [20]. The differences between the reference age and the age at death were summed for all individuals who died before the age of 75. The estimate of PYLL was calculated per 100,000 population aged 0–74.

### 2.3. Ethics

The study was approved by the Vilnius Regional Biomedical Research Ethics Committee (approval no. 2020/1˗1188˗672). Only anonymized data were used in the study.

## 3. Results

According to the diagnostic criteria applied in the current study, the prevalence of epilepsy decreased from 2016 to 2019, although all-cause mortality remained at an average of 45.3 per 1000 PWE (Table 2). Standardized all-cause mortality ratios each year are presented in Figure 2. The frequencies of different causes of death are presented in Figure 3 and Appendix A Appendix A.

Of 6352 deaths documented from 2016 to 2019, epilepsy was the underlying cause of death in 163 (2.6%) cases (average age 51.8 ± 18.4 years), with status epilepticus in 31 (0.5%) cases (average age 51.9 ± 16.7 years). On 1010 (15.9%) death certificates, epilepsy was a contributing cause of death. Epilepsy was either the underlying or contributing cause of death in 1162 (18.3%), with status epilepticus in 44 (0.7%) cases. There were 77 (1.2%) cases in which no clear cause of death could be identified (average age 50.6 ±12.9 years). Over time, there was a statistically significant decrease in external causes of death (r = −0.99, *p* = 0.01) and an increase for the group of endocrine, nutritional and metabolic diseases (r = 0.98, *p* = 0.02), as well as unknown causes of death (r = 0.97, *p* = 0.03).

There were 82 (1.3%) deceased children and adolescents (<18 years old) with epilepsy in our sample (21 in 2016, 2017, 2019 and 19 in 2018, mean age 7.9 ± 5.8 years). The most frequent causes of death in this group were cerebral palsy (25, 30.5%), congenital malformations (15, 18.3%), neoplasms (10, 12.2%), external causes of death (9, 11.0%), epilepsy (7, 8.5%), as well as respiratory and other infections (6, 7.3%).

SMRs for groups of causes of death are presented in Figure 4. SMRs of external and unknown causes of death are presented in Table 3. Of the 102 persons who committed suicide, 96 (94.1%) were male. A mood disorder was documented in 20 (19.6%) individuals with self-inflicted death (15, 75.0% were male).

The most frequent comorbidities among PWE are shown in Figure 5. Only the frequency of cerebrovascular diseases (r = 0.99, *p* = 0.01) and associated para/hemiplegia (r = 1.00, *p* = 0.01) increased with statistical significance from 2016 to 2019. The prevalence of traumatic brain and head injuries varied from 10.2% to 8.1% (r = −0.83, *p* = 0.17), and that of fractures varied from 15.5% to 18.0% (r = 0.18, *p* = 0.82). There were 141 (9.0%) deaths from ischemic heart disease and 254 (8.4%) deaths from any cardiovascular disease (including stroke) with no cardiovascular comorbidity documented during up to three years before death.

## 4. Discussion

### 4.1. All-Cause Mortality

A decrease in the prevalence of epilepsy was observed from 2016 to 2019. Although we are unaware of any legal changes in documenting the diagnosis that could have led to this observation, the trend should be regarded with caution because of a short study period. PWE died around a decade earlier than would be expected for the general population (life expectancy at birth in Lithuania was 76.4 years in 2019—71.5 years for males and 81.0 years for females) [17].

SMRs among PWE remained similar (i.e., around three) from 2016 to 2019, and were within the values reported in other high-income countries [1,13,21,22,23]. However, SMRs are not directly comparable and reflect death rates among PWE only in the context of the local reference population [24]. There were noticeable disparities in mortality between men and women with epilepsy. The predominance of cardiovascular causes of death in the latter group over external causes of death in the former suggest that unnatural deaths among male PWE are a central factor for such inequalities. 

### 4.2. Causes of Death and Comorbidities

The dominant causes of death (cardiovascular diseases, cancer and external causes) reflect the situation of the general population in Lithuania [17].

Epilepsy was documented as a contributing cause of death in around one-sixth of death certificates of deceased PWE, but was noted as the underlying cause of death in only 2.6% of cases. This further suggests that inaccurate choices of the underlying causes of death to be inscribed in death certificates are reflected in administrative databases—this may lead to an underestimation of epilepsy-related mortality among PWE because of death misclassification [25]. For instance, deaths directly attributed to epilepsy in the current dataset were almost sufficient to fulfill the expected count of 1.2 per 1000 patient years of SUDEP cases [7,26]. However, additional instances of SUDEP would certainly be detected among cases of sudden cardiac or unknown death (e.g., if autopsy data were evaluated by experts in the field) and may comprise up to 5% of all deaths [25,27].

The decrease in the proportion of deaths from external causes reflected trends in the general population (a decrease from 7.4% of all deaths in 2016 to 6.3% in 2019) rather than changes specific for the population of PWE—the SMR for deaths from external causes remained constant [18]. The study revealed frequent suicides among PWE; however, mood disorders are only rarely documented in this patient group. Suicides are potentially preventable; therefore, there is a need for better instruments to detect psychological issues and increased suicidal risk in routine settings [28]. Due to the frequent coexistence of epilepsy and social as well as psychiatric determinants of mortality, the prevention of deaths from external causes may be one of the best targets when aiming to reduce premature mortality among PWE [8,9]. Notably, such deaths represent a heterogeneous group of events, many of which are unrelated or unproven to be directly related to seizures (e.g., alcohol, carbon monoxide poisoning, events of undetermined intent), whereas some (e.g., assaults or transport accidents) may be overall less relevant for PWE. In our study, around every four in five of deaths from external causes were among male PWE. It is therefore likely that the incidents are tightly linked to societal factors (e.g., unfavorable living conditions or unemployment) and require interventions by social or community workers in addition to risk-screening in the clinical setting.

The current study suggests that the mortality of PWE is increased across the full spectrum of medical conditions. Some of the latter are primary disorders that lead to both epilepsy and premature death (e.g., brain tumors), but their lethality may not be directly associated with seizures and cannot be prevented by treating epilepsy. The apparent increase in the rate of deaths from cardiovascular conditions may emerge because epilepsy-related deaths are falsely attributed to cardiovascular causes in spite of absent cardiovascular comorbidities [27]. 

Comorbidity data confirmed that cardiovascular disorders and dementia were highly prevalent among PWE before their death. The growing frequency of cerebrovascular comorbidities and para/hemiplegia is set to translate into a further increase in the burden of secondary epileptic disorders [29].

### 4.3. Study Limitations

The limitations of the study consist of a short study period and the use of administrative databases rather than national registries to confirm the diagnosis of epilepsy and its comorbidities. Furthermore, only anonymized data were analyzed and there was no possibility to validate the inclusion protocol of our study. The documentation of deaths from cardiovascular conditions in the absence of respective comorbidities suggests misattribution of the causes of death among PWE. Therefore, the use of data from death certificates may be considered the most important source of bias in our study. There was also no distinction regarding the duration of epilepsy among PWE and no information concerning clinical variables (e.g., antiseizure drugs, seizure frequency). We were unable to associate drug-resistance or seizure type with a greater rate of death—such information would be essential in helping to define the subpopulation of PWE that are most likely to die prematurely in our region. Finally, the absence of autopsy data may lead to an underestimation in epilepsy-related mortality.

## 5. Conclusions

The study reveals that from 2016 to 2019, PWE in Lithuania died around three times as frequently as individuals in the general population. Cardiovascular conditions and cancer were the dominant underlying causes of death. Although the proportion of deaths because of external causes decreased over time, the SMR of these lethal events remained high. This finding, as well as the high prevalence of cerebrovascular conditions and dementia, suggests that mortality associated with underlying causes of epilepsy or its comorbidities is likely to become increasingly relevant in the ageing population of PWE.

## Figures and Tables

**Figure 1 ijerph-18-10512-f001:**
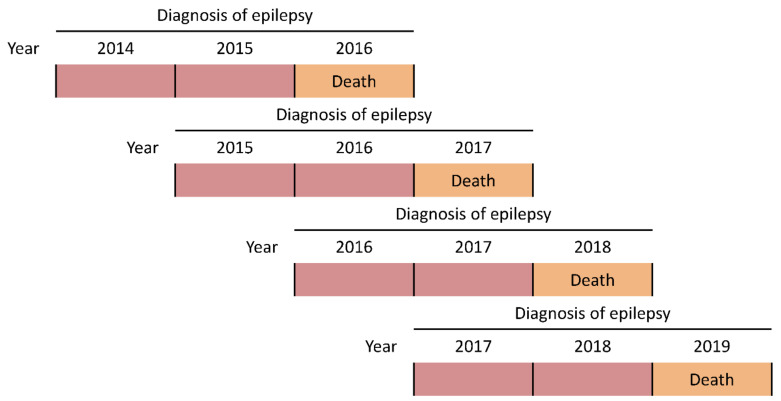
Each year from 2016 to 2019, PWE were included if they were documented to have G40 the year of analysis (orange) and/or anytime two years before (red). The rate of death was compared between PWE and the general population every different year.

**Figure 2 ijerph-18-10512-f002:**
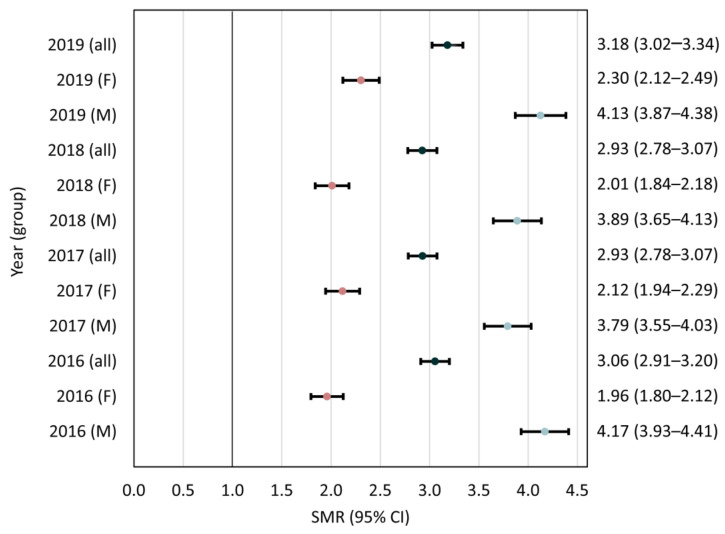
Standardized mortality ratios (SMRs) and 95% confidence intervals (CIs) each year. F—female, M—male.

**Figure 3 ijerph-18-10512-f003:**
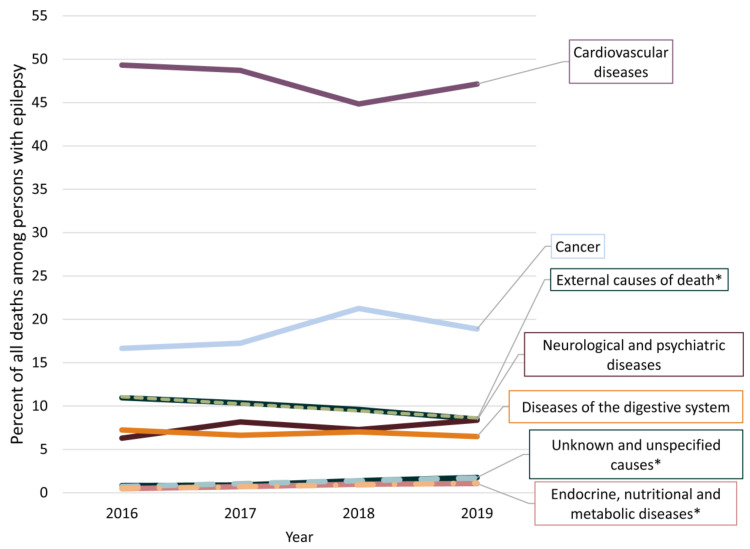
The most frequent causes of death among people with epilepsy every year. Only causes of death surpassing 5% or having trends are represented. * causes of death trending between 2016 and 2019.

**Figure 4 ijerph-18-10512-f004:**
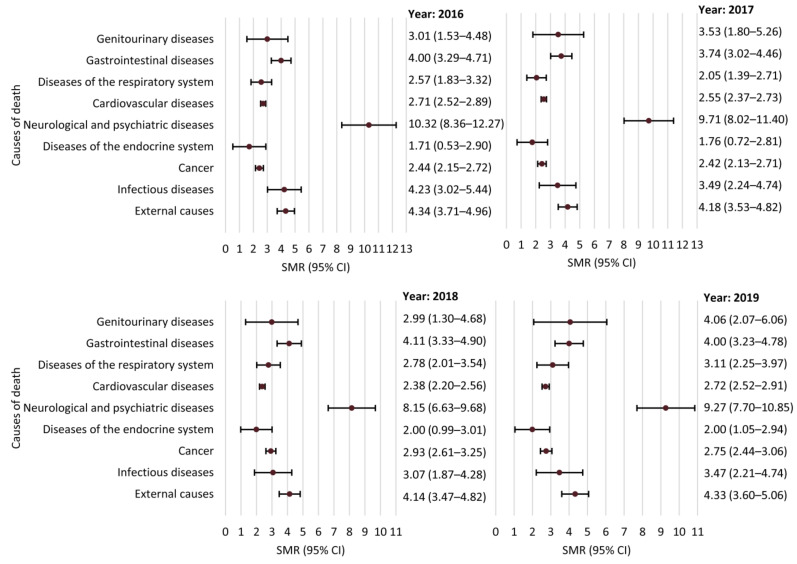
Standardized mortality ratios (SMRs) and 95% confidence intervals (CIs) for different groups of causes of death.

**Figure 5 ijerph-18-10512-f005:**
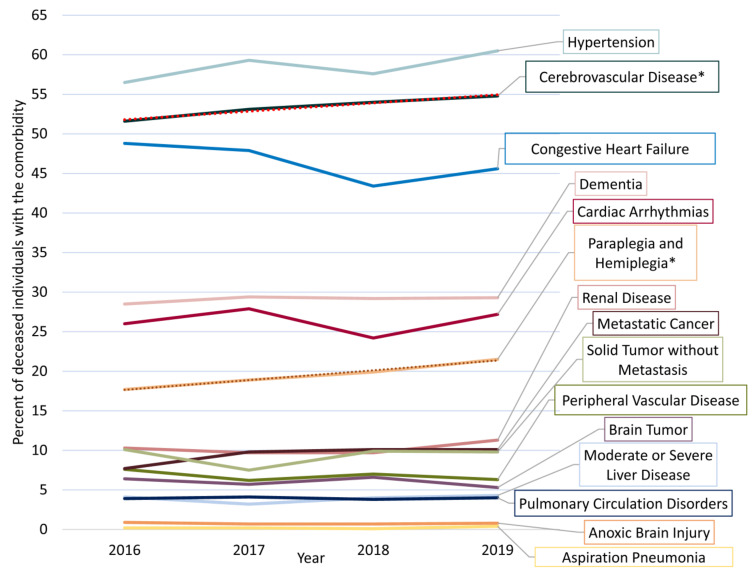
The prevalence of different comorbidities documented among people with epilepsy prior to their death. Conditions comprising the epilepsy-specific comorbidity index as well as cerebrovascular diseases are presented [16]. * comorbidities trending between 2016 and 2019.

**Table 1 ijerph-18-10512-t001:** Data sources used in the study. The study covered a population of 2.8–2.9 million country residents and 33,300–36,700 PWE. ICD-10-AM—the International Statistical Classification of Diseases and Related Health Problems, Tenth Revision, Australian Modification, NCHIF—the National Compulsory Health Insurance Fund.

Data Source	Group	Variables Extracted (for Every Year of Analysis)
**Database of the NCHIF of Lithuania**	People with epilepsy (all)	For every individual: Age; Sex; Diagnosis of epilepsy (ICD-10-AM codes).
**The State Register of Death Cases and Their Causes**	People with epilepsy (deceased)	For every individual: Age; Sex; Diagnosis of epilepsy (ICD-10-AM codes); Comorbid conditions (ICD-10-AM codes); The underlying cause of death (ICD-10-AM code); Causes contributing to death (ICD-10-AM codes).
**The open portal of the National Institute of Hygiene of Lithuania**	General population	Mortality by cause of death and age group in the general population.
**Department of Statistics of Lithuania**	General population	Number of residents in Lithuania by age group; Number of deaths in Lithuania by age group.

**Table 2 ijerph-18-10512-t002:** The prevalence and mortality of persons with epilepsy included in the study. a—Bonferroni-adjusted results of post hoc comparisons for age 2016 < 2019 (*p*_adj_ = 0.020) and 2018 < 2019 (*p*_adj_ = 0.022); b—post hoc result 2018 < 2019 (*p*_adj_ = 0.012); c—expressed per 100,000 individuals aged 0–74; SD, standard deviation.

		**Year**				
		2016	2017	2018	2019	
**Persons with epilepsy**	All	36,746	35,730	34,524	33,252	
	Male	21,729	20,998	20,221	19,326	
	Female	15,017	14,732	14,303	13,926	
	Prevalence (per 1000 inhabitants)	12.7	12.5	12.3	11.9	r = −0.98, *p* = 0.02
**Deaths**	All	1699	1554	1510	1589	
	Male	1150	975	978	992	
	Female	549	579	532	597	
	Mortality (per 1000 PWE)	46.2	43.5	43.7	47.8	r = 0.31, *p* = 0.69
**Age at death ±SD (years)**	All	63.9 ± 17.0	64.5 ± 17.3	63.8 ± 17.0	65.4 ± 17.2	H(3) = 11.465, *p* = 0.009 ^a^
	Male	61.2 ± 15.8	61.6 ± 16.1	61.2 ± 16.0	61.8 ± 16.1	H(3) = 1.979, *p* = 0.577
	Female	69.6 ± 17.9	69.6 ± 18.0	68.6 ± 17.8	71.4 ± 17.3	H(3) = 10.186, *p* = 0.017 ^b^
**Potential years of life lost ^c^**	All	868.3	782.5	799.2	774.8	
	Male	672.4	565.3	586.2	580.6	
	Female	195.9	217.2	213.1	194.2	

**Table 3 ijerph-18-10512-t003:** External and unknown causes of death (standardized mortality ratios and 95% confidence intervals (CIs)) each year. * 95% CI does not cross 1.0.

	**Year**			
	2016	2017	2018	2019
**All external causes of death**	4.34 (3.71–4.96) *	4.18 (3.53–4.82) *	4.14 (3.47–4.82) *	4.33 (3.60–5.06) *
**Transport accident**	3.00 (1.14–4.86) *	2.11 (0.55–3.67)	3.10 (1.08–5.13) *	1.50 (0.03–2.96)
**Drowning**	3.78 (1.44–6.12) *	4.10 (1.26–6.94) *	2.39 (0.30–4.49)	4.55 (1.40–7.70) *
**Fall**	6.41 (4.32–8.51) *	6.47 (4.38–8.55) *	5.86 (3.92–7.80) *	6.11 (4.09–8.13) *
**Poisoning**	4.95 (3.15–6.75) *	3.11 (1.63–4.59) *	4.46 (2.45–6.46) *	4.68 (2.39–6.98) *
**Suicide**	2.96 (1.96–3.95) *	3.11 (2.03–4.19) *	2.09 (1.15–3.03) *	2.02 (1.06–2.97) *
**Event of undetermined intent**	4.71 (2.65–6.78) *	4.84 (2.54–7.14) *	5.70 (3.14–8.26) *	6.75 (3.79–9.70) *
**Assault**	2.91 (0.06–5.76)	1.91 (0–4.56)	5.23 (0.65–9.81)	5.12 (0.10–10.14)
**Unknown or unspecified deaths**	1.84 (0.88–2.81)	1.85 (0.88–2.81)	2.31 (1.32–3.30) *	3.35 (2.11–4.59) *

## Data Availability

Raw study data may be provided by the corresponding author upon reasonable request.

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
