# Peer review of "Mortality among People with Epilepsy: A Retrospective Nationwide Analysis from 2016 to 2019"

_ijerph, 2021, doi:10.3390/ijerph181910512_

Round 1
Reviewer 1 Report
Puteikis and Mameniskiene assessed the dominant causes of death and comorbidities in individuals with epilepsy in Lithuania. This question is particularly timely as Lithuania is one of the countries with a high number of deaths from external causes. The authors show that Cerebrovascular, cardiological, and dementia-related conditions were the most frequent comorbidities among those with epilepsy. Further, epilepsy-related causes of death are relevant and important contributors to mortality amount individuals with epilepsy.
General assessment
The strengths of this manuscript are two-fold. First, the authors devised a well-thought-out analytical framework that can be generalized to examine mortality-related questions in other epilepsy data sets as well as data sets of other neurological conditions. Second, the topic discussed is well-motivated and has important implications for clinical practice as well as basic science research investigating different aspects of epileptic seizures.
Having said this, I found certain information and details concerning the analytical technique to be missing. Moreover, while the result tables are clear and informative, the figures that illustrate the most frequent causes of death and the prevalence of different comorbidities could be substantially improved.
Comments
- The ‘Study design and data sources’ is rather challenging to follow. For example, G40 and G41 should be defined at their first mention to ensure that the article can be understood by a wide range of audience. I also have a hard evaluating how large and detailed the data set is. Some important information regarding the data set is hard to find/seems to be missing. Could the authors consider summarize important aspects of the data set including the total number of (usable) data points and corresponding factors such as age, seizure type, and/or age of onset in applicable/readily available? Overall, I think that the readers could benefit a lot more from the manuscript, if the authors could provide some more information about the data set used.
- Additional information is also needed in ‘Data analysis’. What was the rationale behind using ‘visual inspection of the histogram’? Also, why was Kolmogorov-Smirnov test was chosen to evaluate differences in the data set instead of for example, non-parametric bootstrapping analysis? A sentence or two to provide rationales for using such tests will help better the readers’ understanding of the analytic methods and the results.
- What are ‘padj’ reported in the caption of Table 1? How are these different from p values reported in the table?
- I find Figure 3 and 5 challenging to understand. For figure 3, is it possible to also plot error bars or including markers such as asterisks to indicate significant differences in the figure? This is especially important for Figure 5, since the caption does not provice information about significance levels. In addition, the y-ticks of these two figures should use the same convention e.g., both use % or both use whole numbers without %.
Author Response
We thank the reviewer for the effort and time spent to review our manuscript. We have addressed the issues raised, especially pertaining to the clarity of the Methods and Results sections. Please find attached our reply to the review report.

Reviewer 2 Report
The authors present an interesting study on a topic of utmost importance. The fact that PWE in Lithuania die around 10 years earlier than the general population should be even more emphasised and should also be included in the abstract. Besides, I wonder if it seems possible to calculate the number of lost-life-years due to preventable epilepsy-related deaths.
The authors adequately discuss the limitations of the study caused by retrospective data. I would put even more emphasis on the fact that death certificates might not always reveal the correct cause of death. I think this is the biggest bias of the study. The other bias is that no information about severity of epilepsy, seizure frequency, and factors like this are available. This bias is adressed but should be discussed in an even greater extent in my opinion.
Legend to figure 3 "statistically significant trends": I am not sure if this expression should be used (data are either statistically signifiant OR a trend can be found).
The role of "assaults" in contributing to epilepsy-related deaths should be discussed.
Results "the prevalence of epilepsy decreased from 2016 to 2019": The authors should discuss if there was a real decline or if for example documentation was changed.
Author Response
We thank the reviewer for the time dedicated to suggesting changes to our manuscript. We believe the comments helped to improve different sections of the paper. Please find attached our reply to the review report.

Round 2
Reviewer 1 Report
The authors have addressed all of my comments. I have other questions/concerns. Thank you for the thoughtful revision.